# Allostery Modulates Interactions between Proteasome Core Particles and Regulatory Particles

**DOI:** 10.3390/biom12060764

**Published:** 2022-05-30

**Authors:** Philip Coffino, Yifan Cheng

**Affiliations:** 1Laboratory of Cellular Biophysics, Department of Molecular and Cell Biology, Rockefeller University, New York, NY 10065, USA; 2Department of Biochemistry and Biophysics, University of California San Francisco, San Francisco, CA 94158, USA; 3Howard Hughes Medical Institute, University of California San Francisco, San Francisco, CA 94158, USA

**Keywords:** allostery, proteasome, assembly, AAA+ ATPase, proteolysis, ubiquitin, hybrid

## Abstract

Allostery—regulation at distant sites is a key concept in biology. The proteasome exhibits multiple forms of allosteric regulation. This regulatory communication can span a distance exceeding 100 Ångstroms and can modulate interactions between the two major proteasome modules: its core particle and regulatory complexes. Allostery can further influence the assembly of the core particle with regulatory particles. In this focused review, known and postulated interactions between these proteasome modules are described. Allostery may explain how cells build and maintain diverse populations of proteasome assemblies and can provide opportunities for therapeutic interventions.

## 1. Introduction

Proteasomal complexes are formed by multiple functional components, a two-fold symmetric cylindric 20S core particle (CP) bound at one or both ends by different regulatory particle (RP) activators. The “classic” 26S proteasome consists of a CP bound by the 19S RP; however, cells contain additional structurally and functionally distinct RPs that can join CPs. The formation of fully functional proteasomal complexes is tightly regulated. 

Functional and structural data support the existence of multiple forms of regulation in the proteasome. The 20S CP has a chamber in which proteolysis is enacted, and proteasomal RP activators control access of protein and peptide substrates to the CP. Bidirectional communication between the CP and RP complexes serves complex regulatory functions. In this brief and focused review, we describe known and postulated interactions between these proteasome modules.

In general, allostery describes binding at one site of a protein or multi-subunit complex that changes the energy landscape, thereby, resulting in an alteration in the distribution of the population of accessible conformers, typically with regulatory consequences [1]. Here, we focus on two kinds of effectors that modulate interactions between CP with RP. The first consists of substrates or their components or surrogates: degradation substrates, ubiquitin and ubiquitin-processing enzymes and inhibitors occupying proteolytic active sites. 

In the second, central to the theme of this Special Issue on “The Assembly and Function of Proteasomes in Health and Disease”, RP itself can be an allosteric interactor, transmitting information from one proximal end of the CP to a second distal site of RP binding. It will be argued that this form of allostery may influence how CP populations are allocated to alternate types of RPs and how they influence the assembly and stability of such populations.

Allostery requires the transmission of information from a site of binding of a regulator to a remote site or sites where a response happens. The tetrameric oxygen carrier hemoglobin initially drew attention to these matters. Two observations required understanding [2]. First, hemoglobin binds from one to four oxygens, and the initial binding steps increase its affinity for filling the next open O_2_ binding slot. Binding can be described as cooperative among the sites, not independent. Experimentally, binding isotherms were found to show a sigmoid, i.e., S-shaped, pattern, with an initially shallow concentration dependence on O_2_ concentration, becoming steeper as occupancy increased and then flattening as saturation is approached. 

A different pattern would be expected if binding were not cooperative—a hyperbolic curve, as is observed with myoglobin, a muscle homolog of hemoglobin but a monomer with but one O_2_ binding site. A second observation was that CO_2_ promotes O_2_ unloading from hemoglobin; however, the binding isotherm retains its sigmoid shape. Both cooperative binding of O_2_ and its modulation by CO_2_ can be understood as promoting hemoglobin’s primary function: to load O_2_ in the lungs, unload in the tissue and repeat. Importantly, the site at which O_2_ binds is different from the site of CO_2_ binding. 

That distinction gave rise to a pair of descriptive terms: homotypic and heterotypic allostery. Homotypic refers to regulators that are substrates or primary ligands (e.g., O_2_ for hemoglobin). Heterotypic refers to those (e.g., CO_2_) that change the binding of primary ligands (e.g., O_2_) or, for allosterically regulated enzymes, that modulate the enzymatic conversion of substrates. These concepts, homotypic and heterotypic, while serviceable, are not readily applied to the proteasome, wherein distinctions between primary and secondary actors can be elusive. 

Enzymes were also found to be allosterically modulated. In a commonly observed form of such control, an enzyme that initiates a series of metabolic conversions is downregulated by downstream products of the pathway, a form of feedback that adjusts flux through the pathway. A key and early series of experiments demonstrated that the active site of catalysis by aspartate carbamoyltransferase is structurally distinct from the binding site of the downstream products that inhibit catalysis [3,4]. As for hemoglobin, binding of aspartate substrate to the six catalytic chains of aspartate carbamoyltransferase is cooperative. In this way, small changes of precursor concentration within a defined range can cause large changes in product output.

Allosteric information presumably and, in some classic cases, demonstrably, travels a path of coupled conformational changes, a chain of nudges connecting the site of effector binding to the site of altered functional outcome. Various techniques for demonstrating the presence of such a chain, both computational [1,5] and experimental [6], have been described.

## 2. Proteasome Structure

A brief description of proteasome structure must precede a description of its known forms of allosteric regulation. Crystallographic and high-resolution cryo-electron microscopy (cryoEM) studies have provided information on both the structure and dynamics [7,8,9]. An intact proteasome is assembled by the junction of two kinds of protein complexes, a catalytic chamber for proteolysis and docked to it, a module that can have multiple enzymatic and regulatory functions—termed a regulatory particle (RP). The first of these is the 20S core particle (CP), common to all proteasome subtypes. This will be termed CP, without further elaboration. 

The CP is of cylindric form and is composed of 28 proteins (Figure 1A). Its interior is hollow and harbors sites of proteolysis. The CP is built of a stack of four seven-member rings, with the composition α7:β7:β7:α7. In eukaryotes, each of the seven α proteins is distinct but homologous, and that is true of the seven β proteins as well. The β-ring pair is flanked by the α-rings. Within the β-ring, where destruction of target proteins and peptides is performed, three of the seven β-subunits contain proteolytic active sites, with diverse and broad substrate sequence specificity. Entry to this closed interior space is restricted; consequently, substate specificity is determined by access. 

The α-rings of the CP, positioned symmetrically at each end of the cylinder, contribute to providing that conditional access.

The N-terminal regions of the α-subunits form the gate that controls the access of substrates to the CP. In the CP absent RPs, they form a meshwork that partially occludes the CP entry portal, impairing substrate entry and possibly limiting the exit of the peptide products of degradation [10]. Upon RP docking, the obstructive meshwork is disrupted, and the gate orifice widens [11], thus, clearing the portal. High-resolution structures of 26S proteasomes demonstrate that the α-gate is not fully open in the RP-docked basal state when the substrate is absent but transitions to the fully open state as the proteasome assumes successive conformations associated with substrate engagement, translocation and degradation [12].

An RP can cap one or both CP α-rings, and cells contain both singly and doubly capped forms (Figure 1B–D). The paradigmatic RP is the 19S regulatory particle (19S RP), which is more complex in composition than CP and lacks a symmetric character (Figure 1B). It captures ubiquitinated protein substrates, edits and removes the ubiquitin tags (which mark most substrates for degradation), unfolds and translocates the substrate through an interior RP pore and then through the CP α-gate. The 19S RP is composed of two submodules, referred to as the base and lid. The base has 10 subunits, among these six ATPase component proteins which form a hexameric ring with a central pore, through which translocating substrate threads. The interface between seven-member α-ring and six-member ATPase ring forms the primary interface between CP and 19S RP. 

Chemo-mechanical coupling to ATP hydrolysis provides the motive force for substrate translocation and unfolding. The base also includes ubiquitin receptors. The nine-member lid is positioned asymmetrically with respect to the base and makes some contacts with the CP α ring. It includes deubiquitinases whose activity is coupled to the degradation cycle.

## 3. Beyond the Classic Proteasome

The CP can partner with diverse forms of regulatory particles. In addition to the “classic” 19S RP activator, alternative activators of the α-gate include PA200/BLM10 (human ortholog (PSME4), 11S PA28/PA26 (Figure 1C) in its α–β and γ forms (human orthologs PSME1, PSME2 and PMSE3) (Figure 1D), PI31 (human ortholog PSMF1) and VCP/P97/cdc48 [13]. All these RPs can open the α-gate. The last of those listed is an ATPase with an HbYX C-terminal motif. This motif, as described below, has a specific mechanism of gate opening. 

The biochemical and biologic roles conferred on proteasomes by these alternate caps has been little investigated. To further complicate matters, they can participate in formation of asymmetric hybrid proteasomes, ones with the CP capped at one end by the 19S RP and at the other by one of these alternatives. Cells contain substantial amounts of hybrids [14,15,16,17]. A fourth of the proteasome pool consists of hybrid forms [16]. Hybrids composed of 19S RP:CP:11S produce a pattern of peptide products distinct from those generated by classic 26S symmetric proteasomes [14]. How these various hybrids form, are maintained and function is little known.

## 4. Proteasome Allostery Supports Functional Optimization

Allostery fundamentally describes coordination and communication between distinct sites of a multipart and multifunctional protein, or riboprotein. The proteasome rivals the ribosome in size, compositional complexity, multiplicity of active sites and substrate interactions and its importance in the cellular economy. For both, the cellular consequences of its misregulation are grave. Like the ribosome [18,19,20], proteasomes must coordinate multiple active sites and undergo large scale conformational realignments that optimally position its components for enzymatic processing and to move substrates. 

The expectation that proteasomes utilize allostery is supported by experimental data as described in the following examples. These examples are of two types. The first consists of regulators that are substrates or their components. The second consists of RPs that exhibit cooperative binding to the pair of CP sites where they dock. In the following, we will briefly review varies types of allosteric interactions that regulate functions of proteasome in its assembly and engagement with substrates.

## 5. Allostery by Substrates and Their Components

### 5.1. Protein Substrates

Protein substrates of the proteasome—most typically marked for degradation by conjugation of ubiquitin chain—act as homotypic regulators. During their association, translocation, unfolding and degradation, the 19S RP responds with major conformational changes, both internal and in its relationship with the CP as reviewed in [7]. These acrobatics have been captured by the combined application of cryoEM imaging and FRET analysis. In the basal state of the proteasome, the 19S RP ATPase ring and CP entry pore are imperfectly aligned. 

Upon substrate binding, these become co-axial, facilitating substrate passage to the internal CP sites of proteolysis. The ATPase pore loops that drive translocation reconfigure from a plane to form a spiral. The 19S RP lid rotates with respect to the base; one effect of this is to reposition RPN11, which removes ubiquitin chains, closer to the substrate entering the translocation channel. The α gate of the CP is opened, promoting substrate protein entry and presumably, exit of the peptide products of degradation. Individually and in the aggregate, substrate acts to enforce conformational changes that promote its regulated degradation.

### 5.2. Proteolysis Active Site Inhibitors

Inhibition of proteolytic sites within the CP also allosterically influences its association with 19S RP. The best evidence showing such allosteric effect comes from an experiment in which proteasomes were reconstituted by in vivo incubation of CP and 19S RP [21]. In that experiment, assembly was done in the presence of ATP, with or without further addition of the inhibitor expoxomicin. The inhibitor binds tightly and specifically to the β-ring active sites of proteolysis. Removal of ATP by dialysis (which leaves expoxomicin bound) led to dissociation of the 19S RP/CP complex when expoxomicin was absent; however, the complex was preserved by bound epoxomicin. 

Similar results were observed using another inhibitor, Velcade (bortezimib); in this experiment ATP was hydrolyzed enzymatically by apyrase. Both inhibitors, although structurally different, shared the capacity to protect 26S proteasome integrity against ATP depletion. As confirmation that the stabilizing activity of epoxomicin was conferred by its binding at the CP active sites of protein cleavage, the concentration dependence of the inhibitor was found to be similar for 26S stabilization and inhibition of peptidase activity. 

These data were interpreted as evidence of active regulatory communication from the proteolytic sites within the CP to the CP-19S RP interface. In this view, CP active site occupancy, whether by substrates undergoing degradation or inhibitors occupying the active sites where degradation takes place, stabilizes the protein degradation machinery, thereby, supporting processive protein degradation. 

Structural studies using atomic force microscopy [22] further support the conclusion that proteolysis at CP active sites or occupancy by inhibitors that mimic the catalytic transition state (which include epoxomicin and bortezimib) promotes gate opening. This capacity of competitors to redirect the α termini that engage 19S RP is consistent with their stabilizing effect, as described in [21]. High-resolution cryoEM of proteasomes treated with active site inhibitors [23] provided direct evidence of conformational changes propagated from CP to the top of the lid structure, a distance exceeding 150 Angstroms.

### 5.3. Control by Ubiquitin Chains and Ubiquitin Binding Proteins-USP14/Ubp6

Proteasomes interact actively and reciprocally with ubiquitinated substrates and do so through their ubiquitin chain receptors and deubiquitinases. Among the latter are RPN11 (aforementioned and stoichiometric) and USP14/Ubp6 (respectively mammalian and yeast orthologs), Ubp6 is substoichiometric and, when active, edits ubiquitin chains *en bloc*. Binding of polyubiquitinated proteins promotes α gate opening of 26S proteasomes and enhances their peptidase activity [24,25]. They do so at least in part by association via Ubp6; its association with ubiquitin chains (or ubiquitin aldehyde) activates proteasomes by promoting α gate opening [26], thus, stimulating peptidase activity 2- to 7-fold. 

The capacity of ubiquitin chains to promote α gate opening is clearly an allosteric effect; it remains to be determined whether this interaction is mediated exclusively by Ubp6 or involves other ubiquitin chain receptors. Ubp6 has two modes that inhibit rather than stimulate degradation: enzymatic ubiquitin chain removal, which precludes or interrupts substrate capture and further processing and a non-enzymatic inhibitory effect [27]. Docking of Ubp6 to the proteasome, in turn, regulates Ubp6 function, strongly stimulating its catalytic deubiquitinase activity. Ubp6 binds proteasomes through a series of apparently sequential steps, initiated at Rpn1 and completed by positioning of target ubiquitin at the Ubp6 deubiquitinase active site [28,29]. 

Completion of docking confers on the proteasome a configuration incompetent for processive substrate degradation. After ubiquitin chain cleavage and release, proteasome conformation is presumably released from its paused inhibited state. Structural and mutagenesis data support a mechanism whereby allosteric interactions are propagated from an activation loop of Rpt1 to a series of elements of Ubp6. Through this, Ubp6, through its interactions with both ubiquitin and proteasome, provides a switch that controls the timing of proteasome activity and editing of substrate degradation marks.

### 5.4. Processivity of Degradation

Once engagement and degradation of substrate is initiated, this action is normally completed, cleaving substrate, shorn of ubiquitin chains, to peptides. However, that process can be interrupted, resulting in intermediate products that are only partially degraded [30,31]. Intermediate generation is favored by the presence in a substrate of two features: a folded domain that is highly resistant to unfolding and a nearby sequence tract adjacent to a folded domain that impairs pulling by the translocase and thus impairs unfolding. When the proteasome is challenged by simultaneous engagement with a tightly folded domain and a tract that frustrates the application of force, slippage from the translocase results and, in the extreme case, can cause disengagement, thereby, producing degradation intermediates [31,32]. 

Can this be fully understood as a purely local set of interactions within the translocation/unfoldase apparatus, thereby excluding a role for allosteric control? The data suggest otherwise: the presence of ubiquitin chains promotes processivity [33]. The three ubiquitin chain receptors of the proteasome, Rpn1, Rpn10 and Rpn13, have distinct effects here [34]. More broadly, it would be of interest to investigate the role of other proteasome components in control of processivity (e.g., USP14/Ubp6) and to perform agnostic genetic screens for proteins that control processivity.

### 5.5. Activation by Phosphorylation

There are numerous reports of the effects of phosphorylation on proteasomes; these are generally activating Although the precise nature of these effects remains unclear, some are surely allosteric. Of particular interest is phosphorylation of serine 14 of the Rpn6 protein of the 19S RP. That modification is performed by cyclic AMP kinase and hormones that act through that kinase. Phosphorylation at that site accelerates the degradation of ubiquitinated protein substrates [35]. It will be of interest to determine the mechanism of that activation.

## 6. Allostery in RP/CP Assembly

Cooperative binding of primary ligands, introduced in 1965 as a concept to describe a property of tetrameric hemoglobin [36], has a counterpart in proteasome allostery. As described above, initial site occupancy by O_2_ increases the affinity of the remaining open sites of hemoglobin. This binding site interaction gives rise to the sigmoid shape of the O_2_ binding isotherm: loading at low saturation increase slowly, becomes progressively more efficient and flattens as saturation is approached—a sigmoid or S-shaped curve. Such a sigmoid binding isotherm is a hallmark of homotypic interaction by protein complexes that accommodate multiple substrate interactors. An initial model by Monod, Wyman and Changeux, the MWC model [37] sought to explain the mechanism. 

This postulates two structural states in equilibrium, R of low affinity and T of high affinity. Binding of ligand perturbs the equilibrium, redirecting it from R to T. R and T in this formulation describe not merely biochemical states but conformational states as well, in both of which symmetry of the tetramer is maintained. We thus infer, according to the MWC model, two implications of homotypic allostery: sigmoid binding and conservation of two-state symmetry, toggled between low and high affinity by ligand binding. The MWC model has been elaborated, contradicted, replaced by more complex and realistic models [1] but remains a tractable and simple way to consider allostery. How do data and theory apply to binding of proteasome RPs by the CP?

The experiments to be described made use of Archaea proteasomes. Although these objects are taxonomically very distant, they preserve architectural and functional features of eukaryotic proteasomes, are experimentally tractable and provide results informative of the general properties of the proteasomes of all biologic Kingdoms [38]. The archaeal CP, like the eukaryotic CP, is of the form α7:β7:β7:α7; however, all α and all β proteins are identical. The corresponding RP, termed PAN [39], is also radically simpler, composed of a single protein, six copies of which form an ATPase. This functions as unfoldase and translocase. Thematically, in archaea association of PAN to CP opens its α ring.

Studies were initiated by consideration of the following question. Do 20S proteasomes with two open α gates have greater peptidase activity compared to those with one open? To address this, CP were generated that were asymmetric: one of the two α rings contained a point mutation that precludes the docking of diverse RPs [40]. Thus, homogeneous populations of singly and, using control symmetric CPs, doubly capped proteasomes could be made and their activities compared. The initial results were puzzling. 

The RP tested was PA26, a non-ATPase hexameric 11S cap to which the gate-opening motif of the PAN ATPase (PANc) was artificially grafted. This HbYX motif, positioned at the C terminus of PANc, has the capacity to open the α gate by direct interaction [41,42]. The hybrid PA26-PANc cap conferred peptidase activity that was indistinguishable between the symmetric doubly capped and asymmetric singly capped forms. However, when a similar experiment was done using PA26 devoid of PANc, the results were different: the doubly capped form was about twice as active [40]. 

This led to a hypothesis: A CP with two open gates has twice the peptidase activity of a CP with one gate open, suggesting that peptide gate passage can be rate-limiting for activity. PA26 opens only the local gate to which it binds; however, PA26 additionally equipped with PANc opens both the proximal α gate and, remotely and allosterically, the distal gate as well. This conjecture was strengthened by consideration of the distinct modes of gate opening by PA26-PANc and PA26. 

The interaction of PA26-PANc with the α ring induces a rotation in the α subunits and displaces a reverse-turn loop that stabilizes the open-gate conformation [43]. PA26, which lacks the PANc HbYX motif, does not cause α subunit rotation. Both are gate openers but function through different mechanisms. Importantly, only the former transmits allosteric information to the distal gate.

The hypothesis of remote distal gate opening was tested structurally by direct examination, using cryoEM [40]. In CP singly capped by PA26, the proximal α gate was seen to be open, but the distal gate was closed. In contrast, in CP singly capped by PA26-PANc, both gates were open. Additionally, PAN itself, with its native HbYX motif, also allosterically opened the distal gate. 

Distal gate opening was seen in the presence or absence of peptide substrate; active proteolysis at the β ring sites of peptidase activity is therefore not a condition of α gate allostery. The distance between the proximal α ring site of RP binding and the distal α ring, where the regulatory effect is observed, is about 150 Ångstroms, a remarkable distance for propagation of allostery. That PA26 binding opens the proximal gate but not the distal gate of the archaea CP as independently demonstrated using an asymmetric form of an archaea CP and NMR to assess gate opening [44].

An independent hallmark of allostery, cooperative binding, tested the hypothesis of distal gate opening [40]. CP was mixed with archaea RPs (either PA26-PANc or PA26), in 3- or 4-fold stoichiometric excess of RP/CP, to reconstitute a mixture of uncapped and singly and doubly capped proteasomes. The proportion of these three forms was scored by negative stain EM. Positive cooperativity of binding, stated qualitatively, anticipates a deficit of the singly capped form, the consequence of the initial binding event enhancing the affinity of the remaining open site, compared to the open sites of an uncapped CP. 

Performing an analysis across a broad range of of RP/CP ratios, as would be required to define the full binding isotherm and thereby distinguish hyperbolic from sigmoid binding, is impractical by negative stain EM. However, the null hypothesis, that binding is independent and hyperbolic, not sigmoid, makes a specific and testable prediction, a binomial distribution of the three species: P^2^ for doubly bound CP, 2P (1–P) for singly bound CP and (1–P)^2^ for unbound CP, where P is the probaility of an RP binding to an individual α ring. 

Therefore, the null hypothesis predicting a binomial distribution among uncapped and singly and doubly capped proteasomes can be tested and thus accepted or rejected. That statistical analysis was done using the chi square test of significance. Using PA26-PANc, a binomial distribution was rejected with high probability, demonstrating binding is not independent. It could therefore be concluded that PA26-PANc binding is cooperative. In contrast, when reconstitution was done with PA26 lacking a grafted PANc, an approximate binomial distribution among the three forms was seen, consistent with independent PA26 binding to the two sites. The results of the structural and reconstitution studies thus mutually supported a role of HbYX in propagating the allosteric signal [40].

Cooperative binding of a regulatory complex to CP is not confined to Archaea; 25 years ago cooperative binding of mammalian 19S RP to CP was reported [45]. As described above, 19S RP in subsaturating amounts was mixed with CP, and the fraction of uncapped, singly capped and doubly capped proteasomes was scored by negative stain EM. As in the more recent work with Archaea, loading of RP on CP was found to be cooperative, not independent. Conservation of cooperative binding and communication between the ends of the CP is a property conserved between at least two biologic taxonomic Kingdoms.

What is the path for passing information between α rings? CryoEM analysis has been informative of large-scale structural concomitants of allostery in proteasomes; however,, in this case the path of allosteric transmission and dynamics remains elusive and lies beyond the current limitations of resolution of cryoEM. NMR has had some success in studying allosteric interaction within the CP and may be informative in addressing the question [46]. Rotation induced by HbYX insertion into the α ring likely initiates the chain; however, the subsequent path traverses a long and yet unmapped path.

## 7. Plausible Inferences and Further Speculations

Proteasomes exhibit two classic hallmarks of allostery between CP and RPs bearing the HbYX motif: cooperative binding and distal gate opening. These two aspects can be understood as two guises of one mechanism: A preformed open gate in an α ring confers a higher affinity to an RP than does an α ring with a closed gate, thus, favoring the formation of a proteasomal complex with doubly capped ATPase. The two findings—cooperativity and distal gate opening—may thus be regarded as related manifestations of a common underlying mechanism.

However, what regulatory function does this serve? It biases toward double capping over single. For CP plus PAN RP, little difference is observed in the protein degradation between singly and doubly capped forms [40]. Cooperative binding would simply allocate the 19S RP pool, if limiting, toward double capping. However, hybrid proteasomes are abundant—those with an ATPase translocase RP at one end and a non-ATPase RP at the second. 

Their functions are little understood. Cooperative binding may promote hybrid proteasome formation and stabilization. The diverse complexes that cap the CP may have distinct affinities for the α ring; in a cell, these will compete if α ring sites are limiting. In principle, a preformed open gate is a more inviting landing site for a gate-opening cap. Furthermore, cap affinities may be differentially perturbed for a closed gate versus a distal gate opened by proximal cap binding. To add to the complexity of interaction, some caps are both transmitters and receivers of allostery—those with the HbYX motif—and others merely receivers. These potential interactions offer a rich opportunity for the experimental investigation of how cells allocate proteasome-interacting modules.

Proteasomes are not uniformly distributed in cells [47,48]. Proteasomes in cells do not inhabit the ideal world of the biochemist, one of purified protein functioning and interactions in homogeneous solution. Biochemical investigations of interactions among purified proteasome modules are informative but may be insufficient to fully answer the question of how cells generate and control their diversity of proteasome forms. We will likely need to uncover and understand modulators of hybrid assembly as well as the cellular components that mediate localization.

The ubiquitin–proteasome system has a central role in biologic regulation, and its dysfunction is a prominent feature of certain human diseases [49]. The presence of native allosteric sites implies the possibility of therapeutic interventions in this system—an opportunity that has received attention [50]. A spectrum of inhibitors targeted to proteasome active sites are in clinical use and under investigation [51]. Small molecule drugs that modulate allosteric pathways [52,53] are likely to broaden the therapeutic opportunities for usefully perturbing proteasome assembly and activity.

## Figures and Tables

**Figure 1 biomolecules-12-00764-f001:**
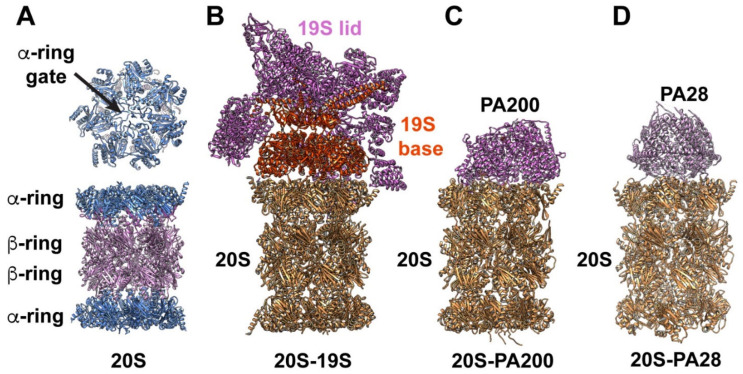
Ribbon diagram illustrating the four major types of proteasomal complexes. (**A**) Top (upper panel) and side (bottom panel) views of 20S proteasome CP. It is formed by a stack of four heptameric rings, two inner β-rings and two outer α-rings. Each ring is formed by seven distinct α- or β-subunits. Proteolytic active sites are contained within the chamber formed by the two β-rings. Top view of the α-ring revealing that, without the activator, the gate (indicated by the arrow) to the proteolytic chamber is closed. (**B**) 20S-19S complex. The base and lid domains of 19S RP are illustrated in different colors. (**C**) 20S-PA200 complex. (**D**) 20S-PA28 complex. In all illustrations, only a single RP is shown docked to the a-ring of 20S CP. In cells, a 20S CP can have two identical or different RPs docked simultaneously to each α-ring, forming diverse types of complexes that are not included in this illustration.

## Data Availability

Not applicable.

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
