# Peer review of "Allostery Modulates Interactions between Proteasome Core Particles and Regulatory Particles"

_biomolecules, 2022, doi:10.3390/biom12060764_

Round 1

Reviewer 1 Report

The authors present a very interesting review of the proteasome through the lens of allosteric changes in different proteasome particles. The authors carefully layout the nature of allostery based on classical observations in any good biochemical textbook. Afterwards they attempt to connect the principles of allostery to the biochemistry and structure of proteasomes, which all though well studied, often dances around issues of allostery or throws the term around indiscriminately and without the care the authors seek to impose on it. For that, I think this is a timely and overdue review. However, mixed within that degree of care there are many oversights and obfuscations that the authors must address before publishing this review.

Line 27: (1) Here, and throughout, the authors use RP (regulatory particles) without clearly indicating whether they are referring to 19S/PA700, 11S/PA28, BLM10/PA200, or p97/VCP. The authors should review their usage of RP in the manuscript and clearly indicate which regulatory particles they are discussing at the moment. (2) Here in the introduction, the authors should enumerate the various RP.

Line 40: At first “multiple types of proteasome assemblages” sounds impressive. Then I realize I don’t really know what the authors mean. Is it the fraction of 20S that is bound by 19S, 11S, PA200, or p97? Or do they mean the breakdown of proteasomes into the different EM classifications of states of proteasomes toward activation? Do they mean the collection on proteasome interacting proteins that modulate proteasome activity? Here, and throughout, the authors favor highfalutin language over precise and clear statements.

Line 77: As this sentence currently reads it sounds as though the α subunits are homologous to β subunits and it isn’t clear if there 7 different α (or β) subunits or just one.

Line 101: Add BLM10 to the alternative names of PA200. The inclusion of PI31 is a bit controversial, because it is not clear yet whether it is an activator. Consider adding the systematic human gene names to PA28 (PSME1, PSME2, PMSE3), PA200 (PSME4), and PI31 (PSMF1).

Line 102: The sentence beginning “The last of those listed …” implies that none of these other RP have a HbYX motif (which is false) and implies that even the 19S RP is not an ATPases.

Line 104: The word choices on this line are a bit odd. “Little investigated” is an unusual phrase and I think that “complicated” should replace “complexify.”

Line 106: It is not clear how much this rather old paper from Keiji Tanaka’s group should be taken as definitive. That paper failed to consider the possibility that a single 19S particle could be on a proteasome, their native PAGE gels also ignore that option. A more modern approach would be the information from cryo-electron tomography such as Wolfgang Baumeister’s group is doing, at least once they can recognize hybrid proteasomes.

Line 109: The section is entitled “Why Proteasome Allostery?” but that question is not answered in this paragraph.

Line 122: Technically ATP is also a substrate of the 26S proteasome. Start, as in line 123, with protein substrates. This section and the following ignores the work of Andreas Matouschek and company defining a two-component element of 26S protein substrates including a sufficiently long and complex loosely folded region and a Ub chain. Biochemical work from Alfred Goldberg’s lab and elaborate cyro-EM structures from several labs upon substrate binding reflect the importance of these two separate components in activation (allosteric changes) but these papers are largely ignored here.

Line 127: The authors do not define the 19S RP N ring. Many readers may be puzzled by this nomenclature.

Line 148: In addition to the excellent work of Maria Gaczynska’s group on atomic force microscopy, there are several rigorous NMR studies by Lewis Kay et al., on this coordination that would really strengthen the argument the authors are trying to make.

Line 153: Although how the inhibited 20S affects the 19S is amazing, the central issue that ties this observation to the thesis of this review is whether changes in the 19S (or 20S gate) can lead to changes in the 20S active site. The authors ought to consider what the implications of the inhibited 20S allosteric changes have for normal allostery in activation. Along such lines, a comment on the proposed allostery within active sites by Alexei Kisselev and Alfred Goldberg (the so called “bite-chew” model) might be useful.

Line 157: A closing parenthesis is missing, presumably after “Ubp6 is substoichiometric.”

Line 158: Is gate opening binary (as suggested by the authors model) or are there degrees of openness (as suggested by Michael Glickman’s yeast mutants).

Line 170: The authors note how ubiquitin binding turns off the inhibitory effects of Ubp6/Usp14 rather nicely. However, they do not address how ubiquitin binding turns on the allosteric activation of proteasomes as described by Alfred Goldberg and Andreas Martin’s groups. Additionally, although it is not an allosteric model per se, the kinetic competition model initially advanced by Dan Finley, is relevant here and should be discussed both as a potential competing model and as one that is not inherently mutually exclusive with the allosteric changes involved with Ubp6. This kinetic competition model will also be useful for developing the processive degradation section that follows.

Line 175: The authors should define what a “nearby sequence tract that impairs pulling” is. I think this would be otherwise cryptic to most readers.

Lines 205 – 208. Several grammatical issues. Line 205 missing comma between archaea and association. Line 206, the sentence ending in the word “question” is a statement not a question – replace the question mark with a period. Line 207: missing comma between “this” and “CP.” Line 208: missing the definite article “the” between “precludes” and “docking.”

Line 222: In the discussion of how PAN versus PA26 activates the 20S gate, it should be commented on how the 19S activates the alpha subunits. Because most structural analysis of the 26S splits proteasomes in half, to increase resolution, they miss this long-range allosteric feature but if the 19S works more like PAN or more like PA26, then we can begin to predict what type of regulation 19S – an inherently interesting form of the proteasome – might have.

Line 230: I am confused here by two points abound sigmoid binding. (1) All graphs in the uncited reference (but presumably the authors own work of Reference 37 in the preceding paragraph) involved hyperbolic curves not sigmoidal curves. Obviously, the authors understand this work better than I do, so think spelling out where exactly this sigmoidal shape will make things clearer. (2) My training in biochemistry focused on the importance of the Hill coefficient rather than the sigmoidal shape (which occurs in many other places not involving protein allostery) as the defining hallmark of allostery. I think the authors may want to more carefully explain what they are describing here.

Line 232: The “null” hypothesis is incorrectly spelled.

Addendum: Many labs have reported the activation of proteasomes by small post-translation modifications, especially phosphorylation. These changes almost certainly involve allosteric changes and makes the understanding of proteasome allostery of real interest to developing small molecule regulators of proteasomes. Therefore, the authors should consider discussing this new area of proteasome biology in their review.

Author Response

Please see the attachment, which is a consolidated response to all three reviews.

Reviewer 2 Report

The present review by Coffino and Cheng focuses on the intricate proteasome machinery offering a valuable attempt to report on the allosteric regulation of the degradation process.

Specifically, the paper addresses the main aspects of the 20S core particle (CP) structure/function/dynamics interplay, with a general insight into the multivariate modulation of proteostasis.

As a multicomponent protein system, proteasome represents one of the major model system whereby allostery could be easily visualized. Multiple proteasomal states have been solved so far, and different structural studies underscored the dynamics and allosteric regulation involved in proteasome assembly.

In this framework, the authors make a huge effort to summarize 20S core particle interactors. Indeed, various examples of allosteric regulation have been uncovered, from regulatory proteins (RP) to substrates and their components. The transition path among different 20S CP conformational states is finely regulated by the concerted interaction with different protein partners.

Accurate substrates degradation relies on an extremely controlled gate entry to the CP catalytic chamber. The cooperative binding of RP to the 20S gate formed by the flexible N-terminal tails of the α-subunits initiates a series of local structural rearrangements underlying the allosteric pathway connecting gating dynamics and substrate translocation and hydrolysis.

A nice example of dynamic long-range communication has been described upon porphyrins cooperative binding to the human 20S α-ring, with particular implications for structural adjustments at the catalyitc sites [see Int J Mol Sci. 2020 Sep 29;21(19):7190].

Overall, the work is well conceived and presented, providing a remarkable amount of information and accessible to a broad readership. The manuscript appears occasionally too verbose, in the revised version the authors should try to condense some repetitive parts, such as those relative to hemoglobin.

In my opinion, there are few minor concerns that could increase the quality of this manuscript and make it suitable for publication in Biomolecules:

1) A more detailed description of the multiprotein assembly from a structural point of view with a mention to the catalytic sites and a clearer discrimination between active and inactive states would be useful. I recommend the addition of an illustrative (even schematic) image of the 20S structure.

2) The authors could evaluate to add an extra paragraph concerning current proteasome-targeting drugs used in clinics, including allosteric compounds, and their implications in cancer and neurodegenerative diseases.

3) A number of typos are present throughout the text. At Page 6, please reformulate the period starting at line 206.

Author Response

(The authors gave the same response as above.)

Reviewer 3 Report

Biomolecules review

Allostery Modulates Interactions Between Proteasome Core Particle and Regulatory Particle

To the authors

The authors made a made a very good work focusing their review on the allosteric modulations that is being identified in the proteasome machinery, especially those which interfere with interactions between the core particle (CP) with regulatory particle (RP). In addition, authors argue that the RP is itself an allosteric modulator of the CP, which binding, therefore, can influence the partitioning among multiple types of proteasome assemblages as well as the assembly and stability of such populations. In my opinion in a so complex theme, full of many excellent published works, as the proteasome structure-function field is, focused revisions are very welcome. I only have two suggestions to make to the authors whose I think would facilitate the understanding of some important points to the readers.

1 – To introduce one or more figures of the proteasome structure showing and indicating the main parts described in the text. The support of those structures would facilitate the understanding of the structural rearrangements and the allosteric modulations so well described in the text, even for readers familiar with some aspects of the proteasome structure-function, and it would be essential for those readers not so familiar with it.

2 – The theme of the review is relevant, but one or more examples could be better described (not only cited) to show (for a more general audience) that this is in fact very important. I would like to suggest using the discussion on the central role that the proteasome impairment seems to have in many neurodegenerative diseases, which may be caused by protein aggregation (Alzheimer, Parkinson, Huntington). It can be discussed and highlighted that the allosteric modulation aiming the restoration of the proteasome activity can be a common treatment for all those diseases (a start point can be the reference: Could a Common Mechanism of Protein Degradation Impairment Underlie Many Neurodegenerative Diseases? By David M Smith. https://doi.org/10.1177/1179069518794675). If authors agree, it can be introduced into the “Plausible Inferences, Further Speculations” topic or even in a final-conclusion remarks topic.

Author Response

(The authors gave the same response as above.)

Round 2

Reviewer 1 Report

The authors have largely addressed the concerns raised in the prior round of reviews and the manuscript reads more clearly though a few troublesome spots remain.

Line 20: Allostery should either influence the assembly of _the_ core particle with regulatory particles (plural) or with _the_ regulatory particle. 

Lines 165 – 184: I disagree with the authors rebuttal that the nature of the signal inducing the allosteric change is irrelevant to their discussion. I had intended to point out an oversight of omission rather than accuse them of failure to know the literature to which they have contributed significantly.

Lines 211 – 213: The authors have not clarified the model of Ub-conjugates binding to Ubp6/Usp14 and thereby activating proteasomes. They may choose to disregard the data of Goldberg et al., and Martin et al.,and posit a model whereby Ub binding to Ubp6/Usp14 de-represses the system. Or they might account for all that data by a system where ubiquitin not only de-represses the inhibition of 26S by Ubp6/Usp14 but also via yet undefined allosteric mechanism activate the proteasome. Certainly, the issue of which model is correct has not been resolved but the authors have raised the problem in their text and should clearly either lay out the resolution with a reasoned argument or posit the two possible models. At the moment, they gloss over that issue entirely and create what appears to be a non-sequitur of moving from activation to inhibition in a puzzling manner.

Lines 298 – 307: The authors have not clarified this experiment, which is really the basis of this review. I am still puzzled by the sigmoidal curve argument because (1) none of the graphs in Yu et al., are sigmoidal – they are all hyperbolic, and (2) there are sigmoidal curves without cooperativity. But the fundamental problem with this section, is given its importance for the purpose of the review is the experiment is not described adequately for one who has not read Yu et al to grasp the basis for the argument (and ideally go and read that fine paper).

Reviewer 3 Report

The authors have made all the additions suggested.

Author Response

Thank you.